# Emergent World Models and Latent Variable Estimation in Chess-Playing Language Models

**Adam Karvonen**
Independent
adam.karvonen@gmail.com

## Abstract

Language models have shown unprecedented capabilities, sparking debate over the source of their performance. Is it merely the outcome of learning syntactic patterns and surface level statistics, or do they extract semantics and a world model from the text? Prior work by Li et al. investigated this by training a GPT model on synthetic, randomly generated Othello games and found that the model learned an internal representation of the board state. We extend this work into the more complex domain of chess, training on real games and investigating our model's internal representations using linear probes and contrastive activations. The model is given no a priori knowledge of the game and is solely trained on next character prediction, yet we find evidence of internal representations of board state. We validate these internal representations by using them to make interventions on the model's activations and edit its internal board state. Unlike Li et al's prior synthetic dataset approach, our analysis finds that the model also learns to estimate latent variables like player skill to better predict the next character. We derive a player skill vector and add it to the model, improving the model's win rate by up to 2.6 times. [1]

## 1 Introduction

Large language models (LLMs) have shown unprecedented capabilities that extend far beyond their initial design for natural language processing. Models trained with a next word prediction task can write code, translate between languages, and solve logic problems. However, these models are also black boxes trained on massive datasets that are typically private. Because of this opacity, there has been debate over how these capabilities emerge. Some have argued that LLMs' apparent competence is deceptive, and is largely due to the models learning spurious surface statistics (Bender et al., 2021). Others have argued that, to an extent, LLMs are developing an internal representation of the world (Bowman, 2023).

One approach to study this problem is by training a general language model architecture on narrow, constrained tasks. Li et al. (2023a) used this approach and trained a language model on the game of Othello. The model OthelloGPT was trained on a next token prediction task on a dataset of Othello game sequences. Despite having no a priori knowledge of the game Othello, the model learned to play legal Othello moves. Using non-linear probes (classifiers that predict board state given the model's activations), the authors were able to recover the model's internal state of the board. They were further able to perform interventions using these probes, by editing the model's internal board state and confirming that it plays legal moves under the new state of the board.

The authors were only able to recover the model's internal board state using non-linear probes, and were unsuccessful when using linear probes. In follow-up work, Nanda et al. (2023) found that the model had a linear representation that could be recovered using linear probes. Nanda et al. were also able to successfully intervene on the model's activations using the linear probes.

---

[1]Code at https://github.com/adamkarvonen/chess_llm_interpretability

However, the authors were only successful when examining a model trained on a synthetic dataset of games uniformly sampled from the Othello game tree. They tried the same techniques on a model trained using games played by humans and had poor results. A natural question arises from this: do language models only form robust world models when trained on large synthetic datasets?

We build on OthelloGPT to investigate this question in a more complex setting by training a language model on chess game transcripts. We find that a language model can also learn to play legal and strategic chess moves, and we train linear probes that recover the internal board state and perform successful interventions using these probes (Figure 1). Thus, we demonstrate that language models can also form world models when trained on non-synthetic datasets.

Our model is trained on real human games, not synthetic games. As a result, there are interesting latent variables we can probe for. We find that to better predict the next character, the model learns to estimate the Elo rating of the players in the game. We validate this representation by using it to both improve and decrease the model's chess playing ability.

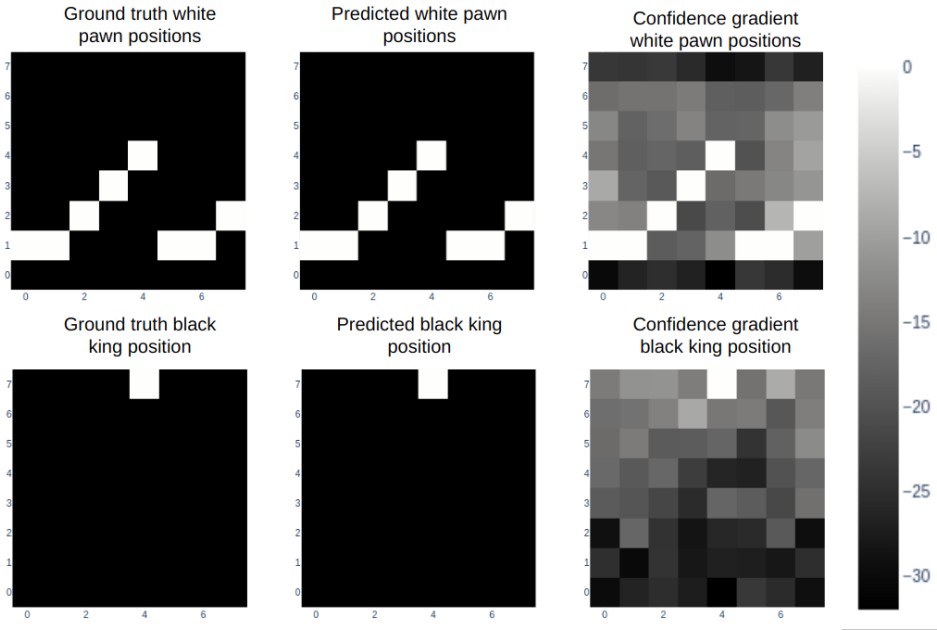

Figure 1: Heat maps of the model's internal board state derived from the probe outputs, which have been trained on a one-hot classification objective. The probes output log probabilities for the 13 different piece types at every square, which we can use to construct a heat map for any piece type. The left heat maps display ground truth piece locations. The right heat maps display a gradient of model confidence on piece locations. To view a more binary heat map, we can clip these values to be between -2 and 0, which can be seen in the center heat map. The model has reasonable representations. It is very confident that the black king is not on the white side of the board.

## 2 Chess Model Training

Our goal is to investigate the internal representations of language models in a more constrained setting. Thus we train a standard language model to play the game, instead of a more targeted system like AlphaZero (Silver et al., 2018), which is built to win competitive games. We intentionally give the model no a priori knowledge of chess to mimic the setting of training on natural language.

## 2.1 Dataset

The authors in Li et al. (2023a) trained two models - one on a 16 million game dataset of synthetic games uniformly sampled from an Othello game tree, and one on 132,921 games from online Othello games played by humans. We hypothesized that the poor performance of the model trained on human games was due to the small size of the dataset. Chess is a much more popular game and there are billions of games available online. We download 16 million games from the public Lichess chess games database, 120 times the size of the original Othello human game dataset.

## 2.2 Model Training

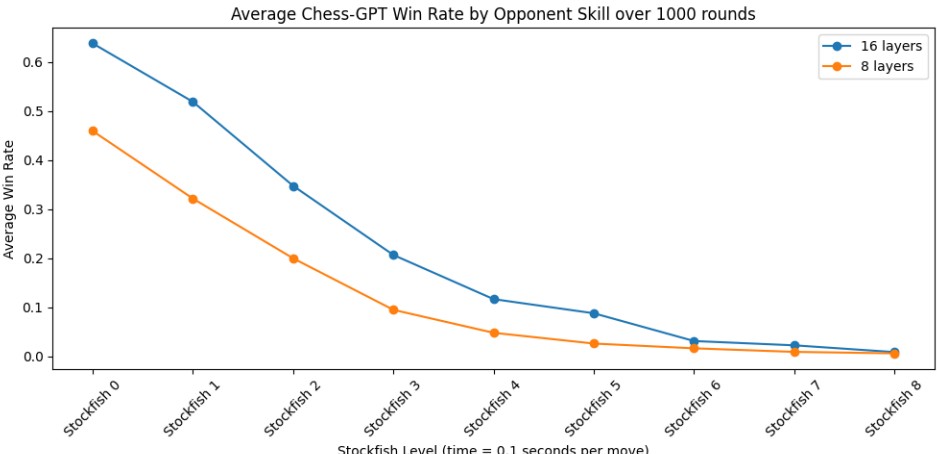

Figure 2: Model win rate versus a range of Stockfish 16 levels. The 16 layer and 8 layer models are trained on identical datasets for an identical number of epochs. The 16 layer model consistently has a higher win rate than the 8 layer model. While precise Elo measurements for Stockfish are complex, a reasonable approximation for level 0 is around 1300 Elo. Details in Appendix E

We train two models, an 8 and a 16 layer GPT model (Radford et al., 2018) with a 512 dimensional hidden space with respective parameter counts of 25 million and 50 million. Each layer consists of 8 attention heads. The input to the model is a chess PGN string (1.e4 e5 2.Nf3 ...) of a maximum length of 1023 characters, with each character representing an input token. The model's vocabulary consists of the 32 characters necessary to construct chess PGN strings.

The model is initialized with random weights and is given no a priori knowledge of the game or its rules. It is solely trained on an autoregressive next token prediction task. For a given sequence of characters, the model must predict the next character.

The models' performance against Stockfish is detailed in Figure 2. The 16 layer model has a legal move rate of 99.8%, and the 8 layer model of 99.6%. After training, the 8 layer model achieved a win rate of 46% against Stockfish 16 level 0 (~1300 Elo), while the 16 layer model had a win rate of 64% (details in Appendix E). Thus a standard GPT, orders of magnitude smaller than modern LLMs like Llama or GPT-4, will learn to play strategic chess if given a dataset of millions of chess games.

We also checked if it was playing unique games not found in its training dataset, rather than regurgitating memorized game transcripts. Because we had access to the training dataset, we could easily examine this question. In a random sample of 100 games, every game was unique and not found in the training dataset by the 10th turn (20 total moves). How is the model successfully playing legal moves? A plausible answer is that it learns to

track the state of the board. To examine this question, we can probe the model's internal representations.

## 3 Probing Internal Model Representations

### 3.1 An Internal Board State Representation

A standard tool for measuring a model's internal knowledge is a linear probe (Alain & Bengio, 2016; Belinkov, 2022). We can take the internal activations of a model as it is predicting the next token, and train a linear model to take the model's activations as inputs and predict board state as output. A linear probe has little capacity and the state of the board is not a linear function of the input. If the linear probe accurately predicts the state of the board, then it demonstrates that the model transforms the input into a linear representation of the board state in its activations.

Given a chess board with 64 squares, we aim to predict the state of each square, $i$, using a linear probe. For each square, the probe is designed to classify it into one of 13 possible states (blank, white or black pawn, rook, bishop, knight, king, queen). The linear probe for square $i$ at layer $l$ of the model is formulated as follows:

$$P_{i,l} = \text{Softmax}(A_l \cdot W_{i,l})$$

Where:

- $P_{i,l}$ is the probability distribution over the 13 classes for square $i$, as predicted by the probe at layer $l$,

- $W_{i,l}$ is a $512 \times 13$ weight matrix for the linear probe associated with square $i$ at layer $l$,

- $A_l$ is the 512-dimensional activation vector from layer $l$ of the model,

- The Softmax function is applied to convert the logits into probabilities.

This method involves training separate linear probes for each of the model's layers and each individual square on the board, where each probe at layer $l$ takes the activations after layer $l$ as input to predict the state of square $i$. We follow Nanda et al. (2023)'s finding that the model represents squares as containing (Mine, Yours, Empty) rather than (Black, White, Empty). To use this objective, for every layer, we must train one group of 64 linear probes for predicting the board state at every white move and a second group of 64 linear probes for every black move, rather than a single group of 64 linear probes for every move.

We train our probes on 10,000 games not found in the model's training dataset. An additional set of 10,000 games, also excluded from the model's training dataset, serves as our held out test set. As detailed in Table 1, the most accurate probe achieves a 99.6% accuracy in classifying the state of each square across the test games. Further insights into the model's performance across different layers are detailed in Figure 3. Most notably, the 8 layer model obtains peak accuracy at layer 6, while the 16 layer model doesn't obtain comparable accuracy until layer 12. Intuitively, one might expect that the most efficient approach would be calculating the board state as soon as possible and then using this information to calculate the next move. Instead, we see the deeper model using twice as many layers to calculate its ultimately more accurate board state.

We can visualize the model's internal representation of the board by creating heat maps. These heat maps were derived from the probe outputs, which had been trained on a one-hot objective to predict whether a chess piece, such the black king, was present on a given square (1 if present, 0 if not). As seen in Figure 1, the model's internal representation reflects reasonable facts about the board state. For example, the internal representation is very confident that the black king is not on the white side of the board.

| Model | Square | Elo |
|---|---|---|
| 16 layer trained | 99.6 | 90.5 |
| 8 layer trained | 99.1 | 88.6 |
| 16 layer randomized | 74.7 | 69.3 |
| 8 layer randomized | 75.0 | 70.2 |

Table 1: Comparison of the best square and skill classification accuracies (%) obtained for the most accurate probe for each model. Appendix F contains F1 scores per piece type.

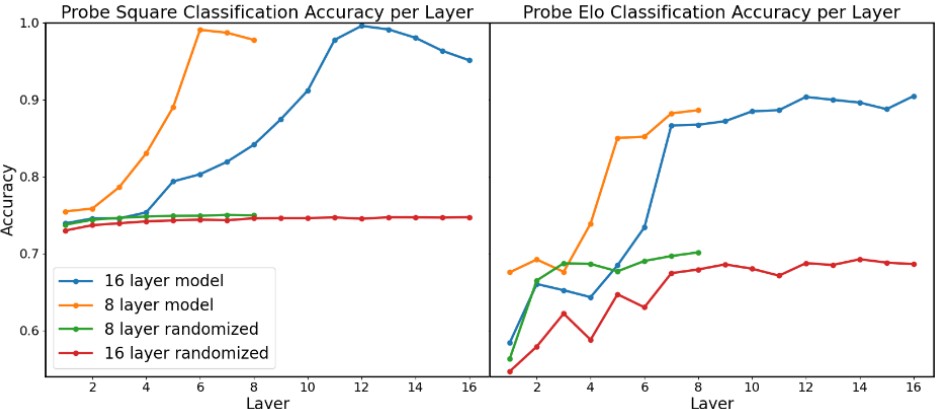

Figure 3: We test linear probes for player Elo classification and board square state classification on every layer of each model. The 8 layer model computes an accurate board state by layer 6, yet the 16 layer model doesn't obtain similar accuracy until layer 12. Oddly, the skill probes trained on randomized models become more accurate on deeper layers.

## 3.2 Probing For Latent Variables

Because the model learned to predict the next move in a competitive game, rather than a game uniformly sampled from a game tree, there are interesting latent variables we can probe for. In particular, our dataset of chess games has the Elo ratings of the players for every game, which we can use to supervise the training of a linear probe.

Initially, we trained the probe on the regression task of predicting the Elo rating of the white player between turns 25 and 35, as it would be difficult to predict player skill early in the game. Despite achieving a promising average error of 150 Elo points, this approach was complicated by the narrow Elo range prevalent in our dataset (the majority of games are between 1550 and 1900 Elo). Distinguishing between players of closely matched skill levels, such as between a 1700 and 1900 Elo player, is intrinsically difficult. This narrow range makes baseline methods appear deceptively effective, as a linear probe trained on a randomly initialized model had an average error of 215 Elo.

To better evaluate the model's understanding of player skill, we trained a probe on a classification task, where it had to classify players as below Elo of 1550 or above 2050. As detailed in Table 1, a probe trained on the real model performs significantly better than a probe trained on a randomly initialized model, suggesting that the trained model learns to estimate player skill in order to better predict the next character. It is further evidence of the ability of neural networks to learn unsupervised representations of high level concepts using next token prediction, especially notable as our models are orders of magnitude smaller than today's state of the art models.

The layer-by-layer performance of our probes is depicted in Figure 3. Peak accuracy for skill classification is obtained at the final layer of the model, unlike board state classification

where peak accuracy is obtained in the mid to late layers. Based on the observed differences in layers for peak accuracy in skill and board state classification, we hypothesize that the model computes a board state and a range of possible next moves in parallel to estimating player skill. At the final layers, the player skill estimate influences which move is selected. The observation that skill probes trained on randomized models become more accurate on deeper layers is peculiar and warrants further investigation into potential unexpected patterns within model architectures.

## 4 Model Interventions

Our probes suggest that the model computes an internal representation of the board. To validate the information found by probes, it is recommended to perform interventions and establish a causal relationship between the internal board state representations and the model outputs (Belinkov, 2022). For our GPT, we use the probes to causally intervene on the model's activations as it is predicting the next token. If there is a causal relationship between the model's internal board state representations and the model's predictions, we should be able to influence the model's chess playing ability and edit its internal state of the board.

### 4.1 Intervention Technique

Linear probes enable a simple intervention approach of vector addition. In this case, our probes are trained on the model's residual stream, which is a 512 dimensional intermediate state vector output by each layer before being fed to the next layer. To intervene on the model's activations, we can simply add or subtract vectors derived from our linear probe from the model's residual stream. Alternatively, we can add or subtract our vector to or from the transformer layer's final Multilayer Perceptron (MLP) bias term, which gives an equivalent intervention that has zero additional inference cost.

$$x' = x + \alpha p_d$$

Here, $d$ represents a specific direction of a probe $p$, such as (High Skill, Low Skill) derived from the skill probe, or one of the 13 different possible square states derived from the board state probe. $\alpha$ is a scaling factor that modulates the intervention's magnitude. $x$ denotes the model's intermediate activations or a layer's final bias term - both 512-dimensional vectors. We can apply this intervention at a single layer or sequentially to a range of layers. For a detailed discussion on the selection of intervention hyperparameters, see Appendix C.

Additionally, we explore the technique of contrastive activations for skill interventions (Rimsky et al., 2023; Zou et al., 2023). In this case, we obtain the average model activation in 2,000 high skill games and 2,000 low skill games between turns 25 and 35. We subtract the low skill activation from the high skill activation to obtain a skill vector, which we can again add or subtract from the model's residual stream or MLP bias term.

### 4.2 Board State Interventions

Our board state intervention is detailed in Figure 4. We test our intervention on 5,000 different board states. Intuitively, we find a piece the model intends to move and remove the piece from the model's internal board state by modifying the model's activations (we do not alter the input PGN string), then sample additional moves from the model. If our intervention is successful, the model's moves will be legal under the new hypothetical board state.

To evaluate our intervention, we sample five probabilistic moves from both the original and modified model at a temperature of 1.0. The original model provides a baseline legal move rate with no intervention applied, and the modified model's legal move rate measures the success of the intervention. Our intervention significantly outperforms our baseline as detailed in Table 2. However, the best intervention success rate obtained is 92%, indicating

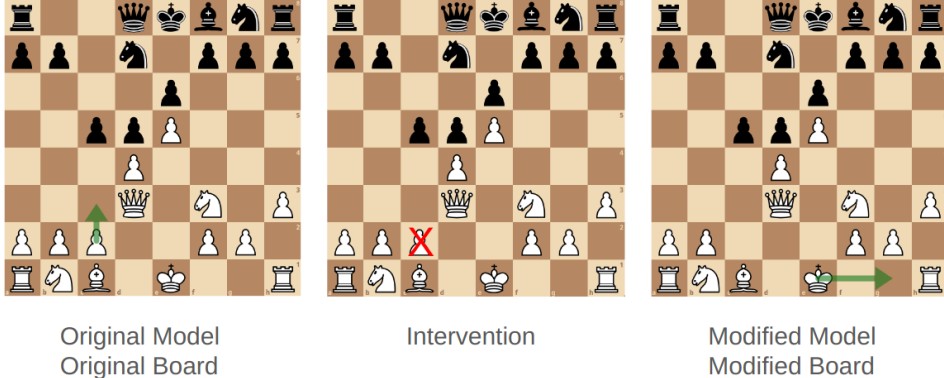

Figure 4: Board State Intervention Process. We first sample the model's next move prediction, identifying a strategically relevant piece the model intends to move (e.g., white pawn from C2 to C3). We then delete this piece from both the original board and the model's internal representation by subtracting the corresponding vector (the 512-dimensional "my pawn" vector from the C2 square linear probe) from the model's residual stream. Using the unmodified PGN string input, we generate 5 new moves from the modified model. A successful intervention results in all moves being legal under the hypothetical modified board state, despite the model receiving no explicit information about the piece removal.

| Model | Original Board | Modified Board |
|---|---|---|
| 16 layer with intervention | 85.4 | 92.0 |
| 8 layer with intervention | 81.8 | 90.4 |
| 16 layer no intervention | 99.9 | 40.5 |
| 8 layer no intervention | 99.8 | 41.0 |

Table 2: Legal move rates (%) for models with and without the board state intervention on the original and hypothetical modified boards (see Figure 4). Results based on 5,000 test cases, sampling 5 moves per case at temperature 1.0. The baseline is the original model on the modified board (bottom right quadrant). This has only 41% legal moves, typically a result of the original model moving the deleted piece. By applying our intervention (top right quadrant), the legal move rate improves significantly to 92%. The original model on the original board is in the bottom left quadrant. The modified model's legal move rate decreases on the original board when it moves pieces into occupied squares (top left quadrant). An example cause would be moving the D3 Queen to C2 in 4. This is legal on the modified board but illegal on the original board.

that there is potential for improved model intervention strategies. As seen in Figure 5, the intervention can have unintended side effects and make the positions of other pieces less distinct.

## 4.3 Model Skill Interventions

We obtain one skill vector from contrastive activations and the second by subtracting the linear probe's low skill vector from its high skill vector. We can add the skill vector to the model's activations to increase its skill. We can flip the sign of the intervention to decrease the model's skill. An excessively large intervention will cause the model to output illegal moves and random characters. When testing our intervention to both increase and decrease model skill, we ensure that the model still makes over 98% legal moves after the intervention. Although the skill probe was trained to predict the Elo ratings of players specifically between turns 25 and 35, we add the skill intervention vector to the model's

| Model and Board | No Intervention | Positive | Negative |
|---|---|---|---|
| 16 layer standard board | 69.6 | 72.3 | 11.9 |
| 8 layer standard board | 49.6 | 54.8 | 5.5 |
| 16 layer random board | 16.7 | 43.2 | 5.9 |
| 8 layer random board | 15.7 | 31.3 | 3.6 |

Table 3: Model win rate (%) against Stockfish level 0 over 5,000 games with positive, negative, and no interventions. We can add (positive intervention) or subtract (negative intervention) the skill vector from the model's activations to increase and decrease its chess playing ability. We test the model on both a standard chess board and on boards initialized with 20 random moves (random board). In the case of the random board, we can improve the model's win rate by up to 2.6 times by adding our positive skill intervention.

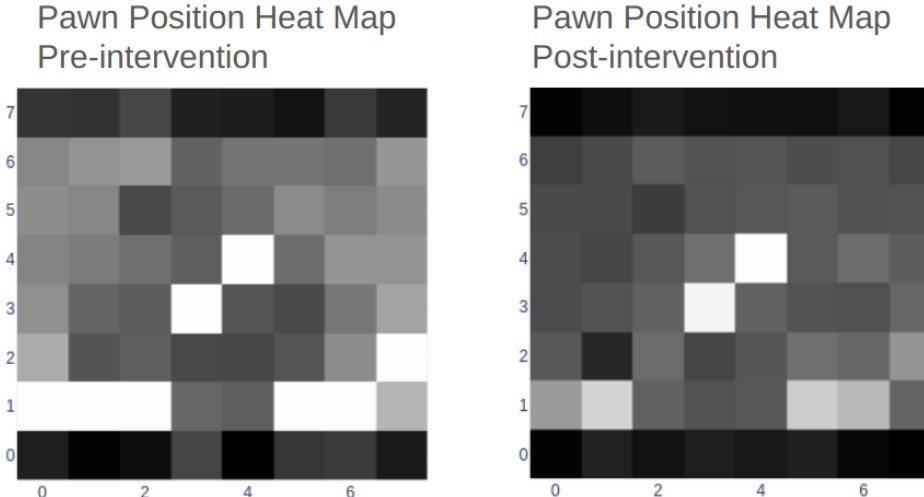

Figure 5: In this intervention, we delete the C2 pawn described in Figure 4. After the intervention, the C2 pawn has been erased from the model's internal representation. However, the other pawns are less distinct, indicating that the intervention has had unintended side effects.

MLP bias term, which means that it is applied at every token in the input PGN string. We observe that it maintains effective performance across all stages of the game.

We perform this intervention in two settings. In the first, we have the model play 5000 games against Stockfish 16 beginning from a standard chess board. In the second, the model plays 5000 games against Stockfish 16, but every board is initialized with 20 randomly selected moves. The results are contained in Table 3. In the standard board, we notice a slight but measurable increase in skill when adding the skill vector, and a significant decrease in skill when subtracting the skill vector. A possible explanation is that the average skill level per game in the model's training dataset is 1,690 Elo, higher than the model's capability and limiting the room for performance improvement in typical scenarios. In the randomly initialized board, both models perform significantly better with a positive skill intervention. The 8 layer model's win rate improves by 2x, and the 16 layer model's win rate improves by 2.6x. The 16 layer model's larger positive increase is possibly because it has more latent ability. More details are contained in Appendix E.

One potential explanation is a model predicting the next character in a game with 20 random moves predicts the next character as if the players have low skill. By adding a positive skill intervention, the model can restore a significant fraction of its ability. To investigate this

hypothesis, we tested our Elo probe on 4,000 randomly initialized games. The Elo probe classified 99.4% of these games as low skill, adding evidence in favor of this explanation.

## 5  Related Work

This work was a follow-on to work done by Li et al. (2023b) and Nanda et al. (2023) where they trained an LLM on synthetically generated games of Othello. Early models such as AlphaZero were trained to play competitive chess and Go, and given built in knowledge of the board and game rules (Silver et al., 2018). In these models, evidence has been found of familiar chess concepts (McGrath et al., 2021). Toshniwal et al. (2021) trained a language model to play chess and found it could play legal moves, although they did not investigate its internal representations.

There have been several examples of researchers finding semantics extraction in Large Language Models trained on text. In 2017, researchers trained a long short-term memory network (LSTM) to predict the next character in Amazon reviews and found that linear probes trained on the model exceeded the state of the art in sentiment classification (Radford et al., 2017). More recently, researchers have found evidence of concepts such as truthfulness and emotion in the Llama LLM (Li et al., 2023b; Zou et al., 2023).

A growing body of work has looked at intervening on language models using these representations, such as by changing the sentiment or truthfulness of the model's response by modifying its activations. Li et al. (2023a) used gradient descent to change Othello-GPT's activations. Other approaches perform linear arithmetic with "truth" vectors Li et al. (2023b), "task vectors" Ilharco et al. (2022), or a variety of steering vectors (Zou et al., 2023; Rimsky et al., 2023; Subramani et al., 2022).

## 6  Conclusion

We provide evidence that an LLM trained on a next token prediction task can develop a world model of complex systems such as chess, including the ability to estimate latent variables such as player skill. We validate these representations by making causal interventions on the model to increase and decrease its skill and edit its internal state of the board.

Chess is a constrained setting compared to the domain of natural language. This constrained setting makes it possible to perform these nuanced measurements and interventions and provide valuable insights into the underlying mechanisms of steering vectors and other concepts in large language models. In addition, it also enables exploration and comparisons of a variety of interpretability and intervention methods. There is room for improvement in our intervention approaches, and we can easily measure our ability to make fine-grained changes to the model's activations, such as by deleting a single piece from the board.

However, chess is ultimately a well-defined, rule-based system that does not capture the complexity and ambiguity present in less structured environments like natural language. Accurate understanding of LLM world models could enable many applications, such as detecting and reducing hallucinations. Our hope is that these techniques can become practical in these domains.

### Acknowledgments

We thank Walt Woods, Alexander Grushin, Martin Wattenberg, Kenneth Li, Nikola Jurkovic, Jannik Brinkman, and Austin Davis for feedback and advice.

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

## A   Hyperparameters

| Hyperparameter | Value |
|---|---|
| Optimizer | AdamW |
| Learning Rate | 0.001 |
| Weight Decay | 0.01 |
| Betas | 0.9, 0.99 |
| Training games | 10,000 |
| Epochs | 5 |

Table 4: Hyperparameters used for training our linear probes.

| Hyperparameter | Value |
|---|---|
| Optimizer | AdamW |
| Learning Rate Schedule | Cosine |
| Max Learning Rate | 3e-4 |
| Min Learning Rate | 3e-5 |
| Weight Decay | 0.1 |
| Betas | 0.9, 0.95 |
| Batch Size | 100 |
| Block Size | 1023 |
| Training Iterations | 600,000 |
| Dataset Characters | 6.5B |
| Dropout | 0.0 |
| d_model | 512 |
| n_heads | 8 |
| n_layers | 8, 16 |
| Parameters | 25M, 50M |

Table 5: Hyperparameters used for training our GPT models.

The 25M parameter model took 72 hours to train on one RTX 3090 GPU. The 50M parameter model took 38 hours to train on four RTX 3090 GPUs.

## B   Reproducibility

All code, models, and datasets are available on GitHub and HuggingFace. The code is located at: https://github.com/adamkarvonen/chess_llm_interpretability

Links to the HuggingFace models and datasets are located in the GitHub repo.

## C   Intervention Technique Comparisons

When performing interventions, there are two primary hyper-parameters: the scale of the intervention and the selection of layers for intervention. In general, we find that the scale parameter should be decreased as the number of layers is increased. In addition, when performing Elo interventions, we had the choice between using vectors derived from the probes and contrastive activations. We observed that the intervention process was successful across a range of parameters, and report the best results obtained for each grouping.

We also explored the following probe based intervention techniques. For board state interventions: both subtracting the moved piece vector and adding the blank square vector. For skill interventions: subtracting the low skill vector XOR adding the high skill vector. All

techniques worked, but the techniques that obtained the best results were detailed in the paper. That is, only subtracting the moved piece vector for board state interventions, and subtracting the low skill vector and adding the high skill vector for skill interventions.

When using probe based interventions, we normalize the probe derived vectors before multiplying them by the scale coefficient. We find slightly better performance by not normalizing for contrastive activation based interventions.

The following are the hyperparameters used for the best result obtained for each configuration. We did not perform an exhaustive hyperparameter search. In all skill intervention tests, we find that contrastive activations are more successful than probe derived interventions.

| Model and Intervention | Layers | Scale | Intervention Type |
|---|---|---|---|
| 8 layer board intervention | 4-7 | 1.5 | Probe derived |
| 16 layer board intervention | range(5,15,2) | 2.0 | Probe derived |
| 8 layer positive Elo intervention | 4-7 | 0.1 | Contrastive Activations |
| 16 layer positive Elo intervention | 11-15 | 0.1 | Contrastive Activations |
| 8 layer negative Elo intervention | 3-7 | -0.2 | Contrastive Activations |
| 16 layer negative Elo intervention | 1-15 | -0.1 | Contrastive Activations |

Table 6: The hyper-parameters used for the most successful intervention in each configuration.

| Model and Intervention | Layers | Scale | Win Rate |
|---|---|---|---|
| 8 layer positive Elo intervention | 5-7 | 0.6 | 24.7 |
| 16 layer positive Elo intervention | 13-15 | 2.5 | 42.2 |
| 8 layer negative Elo intervention | 5-7 | -1.2 | 3.8 |
| 16 layer negative Elo intervention | 7-15 | -1.1 | 7.9 |

Table 7: The best performing probe derived skill intervention win rate (%) and hyperparameters. The positive interventions are performed on random boards, and the negative interventions on standard boards.

We find that a single layer intervention can work almost as well as a multiple layer intervention. We had the following board intervention success rates per single layer intervention.

| Layer | 1 | 2 | 3 | 4 | 5 | 6 | 7 | 8 |
|---|---|---|---|---|---|---|---|---|
| Success Rate | 7.7 | 23.0 | 47.6 | 73.4 | 85.5 | 77.2 | 56.8 | 42.5 |

Table 8: Success rates on single layer board state interventions on our 8 layer model. Notably, the intervention on layer 4 is 85.5% successful, within 5% of the multi-layer intervention success rate. However, this intervention is very sensitive to the choice of layer and is thus less robust than a multi-layer intervention. For single layer interventions, it's crucial to increase the scale parameter compared to multi-layer interventions.

## D  Dynamic Scale Experiments

To estimate the upper bound of the success rate for board state interventions, we explored varying the scale of intervention across a range of values for each move, seeking to identify a scale that resulted in the model generating legal moves. We found a scale that resulted in the model outputting legal moves approximately 98% of the time.

We considered dynamically setting the scale parameter by targeting a specific output logit value for the linear probe of interest. In this case, the target value was another parameter to sweep, and -10 seemed to work well.

Let $s$ represent the scale factor applied to the intervention vector, $A$ denote the model activation vector, $P$ the linear probe vector for the piece of interest, and $V$ the intervention vector. All vectors are 512 dimensional. Our goal was to adjust $s$ such that the linear probe's output for the modified activations equals a predetermined target logit value, $t$. The formula for calculating $s$ is given by:

$$s = \frac{(A \cdot P - t)}{(V \cdot P)}$$

With some algebra, the equation can be rewritten to solve for the target logit value, $t$, as follows:

$$t = A \cdot P - s \cdot (V \cdot P)$$

This calculation ensures that, after applying the scaled intervention $s \cdot V$ to the model activations, the output logit of the linear probe approximates $t$, in this case chosen to be $-10$. It worked well, but was consistently around 1-2% worse than simply using a single scale value.

## E  Model evaluations against Stockfish

For the win rate, a win counts as 1 point, a draw as 0.5, and a loss as 0.

One limitation of our GPT models is the maximum input length of 1,023 characters, which limits the PGN string input to a maximum of approximately 92 turns or 184 moves. After that point, the model can continue to make legal moves if we crop the input to block size, but it doesn't have the full context of the game history. When playing against Stockfish, we stop the game at 90 turns and use Stockfish to evaluate the centipawn advantage at the current board state. If either player has an advantage greater than 100 centipawns, they are assigned the win. Otherwise, it is counted as a draw.

In the case of performing model skill interventions, we limit Stockfish to searching 100,000 nodes per move instead of limiting its search time per move. This uses significantly less search time per move, aiding hyper-parameter sweeps, and reduces variability that may be introduced by performing the comparison on different processors or processors under variable loads.

Elo estimation of Stockfish levels is complex with several variables at play. An official estimate of Stockfish 16 Elos can be obtained in a commit on the official Stockfish repo at commit a08b8d4[2]. In this case, Stockfish 16 level 0 is Elo 1,320. The official Stockfish testing was performed using 120 seconds per move, which is prohibitively expensive for tens of thousands of games on GPU machines. The particular processor being used per machine is another variable.

To estimate the Elo of Stockfish level 0 at 0.1 seconds per move, we had it play 1,000 games as white against Stockfish level 0 with 0.1, 1.0 and 10.0 seconds per move. The win rate of 0.1 vs 0.1 seconds was 51%, 0.1 vs 1.0 seconds was 45%, and 0.1 vs 10.0 seconds was 45%. At low Stockfish levels, a 100x decrease in search time makes little difference in Stockfish's win rate. In contrast, at higher Stockfish levels, decreasing search time makes a significant difference in win rate. Thus, a reasonable approximation of Stockfish 16 level 0 with 0.1 seconds per move is Elo 1300.

## F  Probe F1 Scores by Piece Type

Table 9 provides a detailed breakdown of the linear probe performance for each piece type and board state. The trained models (8-layer and 16-layer) show consistently high F1 scores

---

[2]https://github.com/official-stockfish/Stockfish/commit/a08b8d4

| Piece Type | 8 Layer Model | 16 Layer Model | 8 Layer Random Init | 16 Layer Random Init |
|------------|---------------|----------------|---------------------|----------------------|
| Opp. King | 0.985 | 0.983 | 0.658 | 0.667 |
| Opp. Queen | 0.986 | 0.986 | 0.573 | 0.576 |
| Opp. Rook | 0.973 | 0.987 | 0.638 | 0.647 |
| Opp. Bishop | 0.985 | 0.996 | 0.480 | 0.480 |
| Opp. Knight | 0.986 | 0.993 | 0.494 | 0.473 |
| Opp. Pawn | 0.980 | 0.997 | 0.604 | 0.602 |
| Blank | 0.993 | 0.997 | 0.821 | 0.818 |
| My Pawn | 0.997 | 0.994 | 0.641 | 0.631 |
| My Knight | 0.991 | 0.997 | 0.519 | 0.499 |
| My Bishop | 0.994 | 0.998 | 0.489 | 0.475 |
| My Rook | 0.986 | 0.995 | 0.635 | 0.646 |
| My Queen | 0.992 | 0.998 | 0.569 | 0.570 |
| My King | 0.998 | 0.997 | 0.684 | 0.644 |

Table 9: Linear probe F1 scores for predicting various piece types per model. The probes were evaluated on layer 6 of the 8 layer model and layer 12 of the 16 layer model.

across all piece types, with most scores above 0.98. In contrast, the randomly initialized models show much lower F1 scores, typically in the 0.5-0.6 range, with better performance for blank squares.

