# OpenReview forum: "Emergent World Models and Latent Variable Estimation in Chess-Playing Language Models"
_colmweb.org/COLM/2024/Conference — COLM_

### Official Review · Reviewer_nmzh · 2024-05-02

**Rating:** 7
**Confidence:** 4
**Ethics Flag:** 1

**Summary:**

Builds up on previous works from [Li et al (2023)](https://arxiv.org/pdf/2210.13382) and [Nanda et al. (2023)](https://arxiv.org/pdf/2309.00941) to show that autoregressive transformer decoder models, when trained on chess moves, can learn to capture the board-state in its internal representations (latents). The authors also show that the model is capable of guaging the still of the player that the model is trying to imitate. And it is possible to improve the model's performance in certain settings with a simple intervention by adding a skill vector.

**Questions To Authors:**

1. "... To use this objective, we must train separate linear probes for predicting the board state at every white move and every black move ..." - I was a little lost here. Correct me if I am wrong, but this is how I understand it: A probe $P_{i, l, m}$ is trained for a square $i$, at a layer $l$, for a move $m$; where $m$ is either *white turn* or *black turn*? Or $m \in$ {1st white move, 1st black move, 2nd white move, 2nd black move, ...}?

Either way (or I am missing the point completely) this should be clarified in the next revision. And if it is the latter, I am not sure how it follows the insights from Nanada et al. (2023).

2. Data for training the probes is not balanced - at any time you would expect there to be more blank or pawns on the board. How did you ensure that class imbalance is not skewing the accuracy? Consider also reporting the class-wise accuracy or F1 scores. Maybe also add a confusion matrix reflecting the quality of all the probes (combined for all configs) trained for some of the (best performing) layers.

3. On board state interventions, Li et al (2023)'s strategy using gradient descent is complicated, but it gives you enough flexibility to make sure that the intervention doesn't unintentionally effect the state of other squares. How does the simple addition/subtraction intervention ensure that? On Figure 5 you show that this intervention indeed has unintended side effects. Maybe gradients are the proper way to do this intervention? What's your take on this?

4. Altough not explicitly discussed in the paper, it seemed to me that the model is trying to imitate a player according to the estimated skill level. This reminded me of [Andreas (2022)](https://arxiv.org/abs/2212.01681). Maybe by adding the skill vector you can fool the model into thinking that it is imitating a stronger player, and vice versa. Is this a reasonable assumption?


Some minor questions/comments:

 - Missing reference in the abstract "Li et al"?

 - Whenever you draw chessboards the x-axis should always be ticked from $a$ to $h$. This will help following the discussion.

 - "... do language models only form robust world models when trained on large synthetic datasets?" - I found this question a bit odd. Why the dataset has to be *synthetic*? [Li et al (2023)](https://arxiv.org/pdf/2210.13382) used synthetic data since they didn't have access to enough real data for training. It can be argued that it is the size of the dataset that matters, not whether it is synthetic or not.

 - Board state interventions, Table 2 => does the *no intevention - Modified Board* mean that you didn't do an intervention yet, but you are evaluating the legality of next moves on the board state you intend to achieve after the intervention?

 - A glossary of terms/phrases such as; "Elo Rating", "Stockfish" would be helpful.

**EDIT**: Rating increased to **7** after rebuttal

**Reasons To Accept:**

* Provides further evidence for the claim that large transformer models do not just memorize surface-level patterns but also capture the world-model in their latents and use it to make certain decisions.

* Well written paper with strong experiments.

**Reasons To Reject:**

The paper is incremental in nature, directly building up on previous works. A *task* vector that makes the model perform a certain task, is also discussed in previously in the works ([Li et al (2023)](https://arxiv.org/abs/2306.03341), [Illharco et al (2023)](https://arxiv.org/abs/2212.04089), [Todd et al (2023)](https://arxiv.org/pdf/2310.15213)).

---

> ### Author Rebuttal · Authors · 2024-05-30
>
> Thank you for your thorough review and insightful comments.
>
> **Incremental nature**
>
> Our research contributes to the debate surrounding the robust world models found in OthelloGPT. OthelloGPT only had a robust world model when trained on a large synthetic dataset and did not find a robust world model when trained on human games, and there was debate on if this was due to the size or synthetic nature of the dataset.
>
> We emphasize the significance of using chess as a testbed for studying steering vector mechanisms, as chess provides measurable ground truth unlike natural language. Mechanistic interpretability is severely hampered by the lack of a ground truth in language models. We aim to further this promising direction.
>
>
> **Question on linear probes.**
>
>  We agree that clarification is needed. We propose modifying the paragraph as follows:
> "To use this objective, for every layer, we must train one group of 64 linear probes for predicting the board state at every white move and a second group of 64 linear probes at every black move, rather than a single group of 64 linear probes at every move."
>
> **F1 score for probe outputs** We agree that measuring F1 score on a per-class basis would be a valuable addition to the paper. Although we cannot submit revisions during the discussion process, we have ran the experiments and will incorporate this analysis in the final version of the paper.
>
>
> **Gradient descent** We appreciate your thoughts on the gradient descent approach used by Li et al. (2023). In our initial experiments, we found that linear probe addition outperformed gradient descent by several percent, similar to Nanda et al's results.
>
> **Skill vector effects** Your interpretation of the model imitating a player according to the estimated skill level is spot-on. In the case of a game initialized with random moves, a good next-token predictor would predict legal random moves, which is what we find. By adding the skill vector, we can fool the model to imitate a stronger player, even if the game begins with random moves. We appreciate the connection to Andreas.
>
> **Board state interventions, Table 2, no intevention - Modified Board**
>
>  Your understanding is correct. No intervention - modified board serves as a baseline in this case. We are evaluating the original model on the make-believe modified board, which has one piece deleted from it. In this case, 60% of the original model’s moves are illegal because it is attempting to move a deleted piece.

---

> > ### Comment · Reviewer_nmzh · 2024-06-02
> >
> > I thank the authors for their detailed response. And I do appreciate the effort to compliment and extend previous works in this line of research.
> >
> > On board state interventions, as far as I understand, the motivation of this experiment was to check if the model refers to its internal representation of the world (board state) to make decisions. Not to squeeze out more performance with a (dare I say) *"less principled"* intervention.
> >
> > I am satisfied with the authors' response to other questions. I am happy to increase my score to 7. Although it seemed like this paper was gonna get accepted anyways,
> >
> > Congratulations on your fine work!

---

### Official Review · Reviewer_oZhN · 2024-05-11

**Rating:** 9
**Confidence:** 5
**Ethics Flag:** 1

**Summary:**

This study revisits the debate about whether language models (LM) extract semantics and a world model from text, or only surface level statistics through learning to play chess games. Although previous work shows that LM can only learn a world model from synthetic data, the authors hypothesize that the root cause is from the small amount of human behavior data being used. The authors empirically prove this hypothesis by gathering a large amount of online chess game records played by humans, and train a LM to demonstrate it. The authors further attest this by conducting linear probing based on the LM’s internal activations to predict chess board states and player skills, which clearly aligns with the hypothesis.

**Questions To Authors:**

1. P5: for predicting the Elo, the current experiment uses the first 25-35 turns. Do you have studies on the difference between using less turns and more turns in this experiment? Also, why is the number of turns a range here, rather than a constant number?
2. Figure 3: The hypothesis to explain the layer performance difference is that the model learns the chess board state first and then be able to infer the player Elo and next move. How can we design an experiment to dive deeper into this hypothesis?
3. Figure 3: looking at the 16-layer model result, it seems that the model fully understands the chess board at 12th layer, while has a big performance jump in elo classification at 7th layer. Any potential hypothesis to explain this observation?

**Reasons To Accept:**

1. The paper is well written, high-quality, and easy to follow.
2. The experiment design and results are well-positioned and convincing, rendering little space to challenge the idea.
3. The paper has a significant contribution to this line of work by clearly demonstrating that what is learned by LM is beyond surface statistics.

**Reasons To Reject:**

1. A few analyses can be added to support the claim. Check my questions.

---

> ### Author Rebuttal · Authors · 2024-05-30
>
> Thank you for your positive feedback and insightful questions. We appreciate your recognition of the paper's quality, clarity, and significant contribution to understanding what language models learn beyond surface-level statistics.
>
> **Elo probe range**
>
> We chose the range of 25-35 turns as it was far enough along in the game to have sufficient information from which player skill could be inferred. If we use all turns (such as turns 0-35), the linear probe's accuracy decreases. This is because there is little information in early game moves that can reliably predict player skill. For example, it is nearly impossible to predict player skill after the first turn. If we use fewer turns or a constant number, such as only using data from turn 25, we would need approximately 10 times as many games for the linear probe to obtain the same amount of training data as using the 25-35 turn range.
>
> **Layer performance difference**
>
> One hypothesis is that the model computes a board state and probability distribution over next moves, and then uses its skill estimate to shift the probability distribution. In appendix C, skill interventions are typically most successful in later layers compared to board state interventions, offering tentative evidence towards this hypothesis. A more careful examination of intervention success by intervention type and layer number would be helpful. It would be best to use mechanistic interpretability techniques to further understand how the skill estimation and board state interact, but we leave this to future work.
>
> **Figure 3, jump at layer 7**
>
> Our hypothesis is that the model has to process the full sequence of moves and game history in order to compute the board state. It may be able to first estimate player Elo while processing the game history, before board state is fully computed. We notice that in the 8-layer model result, it has a jump in elo classification at layer 5, before peak board state accuracy is achieved at layer 6.
>
> Thank you again for your valuable feedback.

---

> > ### Comment · Reviewer_oZhN · 2024-06-07
> >
> > make sense to me. Thanks for the clarification.

---

### Official Review · Reviewer_cUDn · 2024-05-13

**Rating:** 7
**Confidence:** 3
**Ethics Flag:** 1

**Summary:**

The authors study the ability of a chess playing language model to encode board state and player skill level.  The paper includes layer-wise linear probing experiments, finding that both 8- and 16-layer models can predict board state very accurately at an intermediate layer and player skill classification with increasing accuracy in each layer.  The paper also includes linear interventions, finding that the quality of play can be increased/decreased by adding/subtracting "skill vectors".

This work extends earlier analyses of the capabilities of chess language models, and also adds to the literature on world modeling in language models (at least in highly structured "worlds") more broadly.  Although Toshniwal et al. 2022 showed some world state tracking abilities of chess LMs (without probing the models layers), and Li et al. 2023 and Nanda et al. 2023 performed layer-wise probing experiments on Othello-playing LMs, I believe this paper provides the first analyses of layer-wise behavior of chess models.  I find the results about player skill to be particularly interesting; the ability to judge and adjust player skill is less intuitive than board state tracking, which had already been studied previously.  The paper is generally well written.  The contribution is somewhat limited since only a single model (of two sizes) was studied, and since the chess task itself is naturally limited.

**Questions To Authors:**

- I didn't entirely follow the board state intervention experiment.  I wasn't quite sure how the board is modified, or what is the difference between the board intervention (original vs. modified board) and the model intervention.  I can imagine how it may have been done, but could you provide more details?

- In the case of board state intervention, it seems that the modified board state could be an impossible state.  Is that correct?  If so, can you comment on how this might impact the results?

- A more expansive discussion/conclusion would be helpful, describing how the authors think their work could impact a broader range of models/settings.

Minor wording issues:

- I don't quite follow "the model had a linear representation that could be recovered using linear probes."  Could you clarify?
- "In 2017, researchers trained a long short-term memory ..." -- this sentence doesn't quite parse.  Maybe it's missing "and" after "reviews"?

**Reasons To Accept:**

+ Interesting results
+ Generally well written

**Reasons To Reject:**

- Limited setting
- Unclear implications for other models

---

> ### Author Rebuttal · Authors · 2024-05-30
>
> Thank you for your insightful review and positive assessment of our paper. We appreciate your feedback and the opportunity to clarify a few points.
>
> **Implications for other models**
>
> Crafting steering vectors as demonstrated in the paper has broad implications for the NLP community. Steering vectors are interesting and have powerful effects on LLM behavior, such as increasing truthfulness or reducing hallucinations, but it is difficult to mechanistically study how they are modifying the LLM’s behavior as there is no underlying measurable ground truth in natural language. This is why we believe chess could serve as a valuable testbed for studying the mechanistic interpretability of steering vectors in LLMs and for mechanistic interpretability in general. In chess, there is an underlying ground truth we can measure for the world state, and Stockfish can be used as an oracle to grade the quality of moves after an intervention. The existence of this ground truth makes it much more tractable to understand what is going on internally.
>
> **Clarification of board state intervention**
>
> Intuitively, for the board state intervention, we edit the model’s internal board state (we do not alter the input PGN string) and if our intervention is successful, the model’s move will be legal under the new make-believe board state.
>
> We begin with the original model’s deterministic next move based on a fixed PGN string. We identify the piece P the model intends to move during this first forward pass. Next, we perform another forward pass through the model, but this time we subtract the steering vector for P from the model’s activations. Conceptually, this deletes the piece from the model’s internal world representation. We sample from this intervened forward pass for 5 subsequent moves. If successful, the five moves should all be legal despite our intervention creating a make-believe state within the model’s internal world representation. Finally, we check that the moves are all legal using a separate chess library which we initialized from the FEN representation of the make-believe board where P was removed.
>
> When we delete the piece P above, we check that the modified board is still valid and has at least one legal move available before sampling from the model.
>
> Thank you for pointing out the minor wording issues. We will clarify the linear representation sentence and revise the 2017 research sentence.

---

> > ### Comment · Reviewer_cUDn · 2024-06-05
> >
> > Thank you for the clarifications, and congrats on your interesting work!

---

### Official Review · Reviewer_mgGk · 2024-05-13

**Rating:** 5
**Confidence:** 4
**Ethics Flag:** 1

**Summary:**

### Summary of Paper
This paper shows that a model trained on human chess gameplay learns to track position information and models player skill level (ELO). Causal experiments suggest that these found representations play a part in how the model makes its decisions.

### Summary of Review
Overall, I think this work requires 1) additional experiments, 2) further depth for existing sections, and 3) some clarifications of current experiments and what the results imply. For example, they find that causally intervening on a skill vector can improve performance, but its unclear how that intervention is actually doing this or qualitatively what that change in behavior amounts to. As is, I think its hard to know what to conclude.

**Questions To Authors:**

### Questions
* How do the moves change when you increase the "elo" for the standard-board games? A qualitative or expert or in-depth analysis of a game on a standard board (with the model boosted to perform even better, getting +3% to +5% win rate, and comparing the moves before and after might (while potentially misleading), would be exciting.
* How do move probabilities and rankings change given the ELO/intervened-ELO?
* How good at predicting the lichess games did the models get? Did the error correlate with game ELO? I guess at some point low-ELO games are harder to predict than mid/high level games. For the first, the moves are more random; for the second, its "harder" to find the good moves the players find.
* How do you remove piece (pawn at c2?) and have a valid PGN/board-state?
* In table 2, bottom-right quadrant: This result says that giving the modified board to the model w/o intervention has a high error rate? What makes the model error here?
* In table 2, top-left quadrant: does this mean that 85.4% of the time it picks a new non-ablated piece to move? Does it ever try to move the ablated piece? What happens in that remaining 10%?
* Does the ELO probe work for ranges that it wasn't trained on?
* Table 3: If you subtract random noise of equal magnitude to the elo-vector does the model reach a similar win-rate (11.9)?
* Do you have results for probing for "my pieces"/"your pieces" instead of black/white? [Following https://arxiv.org/pdf/2309.0094]

### Notes
* I find the move-board-state intervention logic confusing; the point is that you modify the state and then board and check that the model's move post the modified board and intervention state are the same? Writing more explicitly what you expect and what the performances mean would be helpful.
* (suggestion) Adding board positions or labels to the axes of Figure 5 would help the reader track what C2 means. You could also add additional manual highlighting to the squares.
* (nitpick) "it is recommended" in the first paragraph of "4 Model Interventions" sounds funny. I'd say, "Following Belnikov, 2022, we..." or directly argue for your case.

**Reasons To Accept:**

* The ELO intervention finding is interesting.
* With some clarification, the fact that interventions are all relatively simpl/linear (but still effective) is telling.
* This work supports a growing body of research, replicating findings observed for othello

**Reasons To Reject:**

### Overview
* Most of the papers' results would be the assumption given prior work (mcgrath and other probing over game papers like tomlin/forde et al + othello gpt, showing that models track game state.) Some readers would be unsure what to takeaway from reading this paper.
* The paper doesn't uncover more on how such models are working under the hood; The exception here is that they do raise the possibility that the model would try to track multiple possible actions and then use the predicted player skill level to choose an appropriate move. Further testing this hypothesis would make this work stronger.
* There are times in the paper where the setup is unclear. The hypothesis/alternatives are not always clearly defined nor how to interpret the results/metrics.

### What to conclude?
It is hard to know what to conclude from reading this paper.

W.r.t. to the positions of the pieces, I think the field understood that a model would track this information. Table 2 (for me) doesn't adjudicate on whether the found representations capture something meaningful. Spelling out the significance of the experiments and their conclusions would help. I think that the results say that 85.4% of the time the ablated piece is successfully removed and ignored simply by a linear intervention. Is that the case? I'm not perfectly confident that is how legal move rates should be interpreted here.

W.r.t. to ELO, I think this is the most exciting part of the paper. Pushing this forward with more study and analysis would make this paper much stronger. (One could almost entirely for-go the position probing, or push it to the appendix.) An in-depth qualitative study of one (a few?) games and how the elo intervention changes the model's specific choices would be illuminating. How does the ranking of moves change with measured ELO? How does the ranking of moves change with intervened ELO?

### Reviewing Dimensions of the Work
Okay ``Empiricism, Data, and Evaluation``

Okay ``Understanding Depth, Principled Approach``
* I think this paper requires more depth; more diving into the details and further evaluation. Ultimately, this approach is empirical and I think empirically evaluating and understanding what is going on would be helpful. See some of my questions/notes for specifics.

Okay ``Clarity, Honesty, and Trust``
* Though I found some parts unclear, I think they positioned their findings with honesty/trust-worthiness.

Limited ``Ambition, Vision, Forward-outlook``

Limited ``Technological Impact``
* Overall, this paper studied previously explored domains (chess board representations) without making a large contribution to further understanding that domain (chess board representations); I think their work does sketch some interesting hypotheses, but require further follow-up experiments.

---

> ### Author Rebuttal · Authors · 2024-05-30
>
> Thank you for the thorough review and feedback. We have addressed your key points below.
>
> **Significance of Results**
>
> Our research contributes to the debate on robust world models in OthelloGPT, which only emerged with large synthetic datasets. There were questions if this was due to size or synthetic nature of the data.
>
> We emphasize the significance of the ELO intervention, demonstrating chess as a testbed for studying steering vector mechanisms, which are also applicable to latent variables like personality and style in LLMs. Chess provides measurable ground truth unlike natural language, making study of these mechanisms tractable.
>
> While OthelloGPT and ChessGPT can track board state, the how is not understood. ChessGPT enables deeper study with greater strategic complexity while maintaining an available ground truth. In general, interpretability is greatly hampered by lack of an existing ground truth in natural language.
>
> **Elo Probe Range**
> The Elo probe works outside its training range. In 4.3, it accurately classifies games with 20 random initial moves as low skill on turn 10 (below the 25-35 training range).
>
> **Board State Intervention**
>
> We edit the model's internal state (not PGN input). Thus the input PGN is invalid relative to the desired board state. If successful, the model's move is legal under the new "make-believe" state.
>
> We subtract the steering vector for piece P to "delete" it in the model's internal representation. We sample 5 subsequent moves, which should be legal under the “make-believe” board. We verify that the modified board is valid and has at least one legal move before intervening.
>
> **Table 2 bottom-right** The original model tries to move the deleted piece, serving as an intervention effectiveness baseline.
>
> **Table 2 top-left** 92% of the time a non-ablated piece is selected. Most errors are attempts to move the ablated piece. The 85.4% quadrant errors are from trying to move a piece into the "empty" ablated square, which is not empty on the original board.
>
> **Addressing Other Questions**
>
> - We represent squares as (Mine, Yours, Empty) per Nanda+2023, training separate probe groups from each player's perspective. We will clarify this.
> - Detailed analysis of Elo intervention impact would be valuable but is not possible during the discussion phase due to conference review guidelines.
> - We did not measure per-move prediction accuracy. The character-level tokenizer loss cannot be straightforwardly converted to move accuracy.

---

> > ### Comment · Reviewer_mgGk · 2024-06-03
> >
> > Thanks for the response and clarifications! I'm excited by your findings and to see where this work goes next, and I'm glad that other reviews were quite positive. As raised, your experiments and findings come through. Though I stand by my comments, I think you make a good point highlighting that previous work was done in a synthetic setting. I am raising my score to a to 5: basically I would suggest some revision for clarity and additional experiments to confirm/understand how the elo intervention changes which moves are made. It looks like your work will be accepted regardless (congratulations!)!

---

### Decision · Program_Chairs · 2024-07-10

**Decision:**

Accept

**Comment:**

Reviewers are enthusiastic about this work.

Strengths: The consensus is that this analysis of the linear representations within chess-game-trained LM transformers is interesting and clearly presented, with solid experimental findings that are a welcome complement and extension to previous similar work on OthelloGPT. The submission differs from that previous Othello work since Chess is different (and more complex) game; unlike the previous work, the probes here are able to achieve high accuracy on models trained on real gameplay data instead of synthetic data. Reviewers appreciated the experiment design including the causal steering-vector interventions for board state, and they highlighted the novelty and significance of the finding of an ELO skill latent which one reviewer described as the "most exciting" finding of the paper.

Weaknesses: The main weaknesses identified by reviewers indicate the reviewers' wish that the paper explored its findings in even more depth, for example, suggesting further analysis such as expert human analysis of elo interventions. And further analysis of the mechanisms that connect probed states to behavior.

Summary - Chess is one of the most venerable and well-studied classic testbeds for AI, and this paper's development of Chess as a testbed and demonstration for transformer LM interpretation methods will be a real value to the community. The authors should consider adding more in-depth analyses such as those suggested by the reviewers; a full-fleshed analysis that includes human evaluation and a few deeper investigations could make this solid paper into a potentially seminal paper.